

# A new miniMOS tool kit capable of visualizing single copy insertion in *C. elegans*

Jiangyun Li[1,*], Yuang Qin[1,*], Chengchen Shen[1], Jun Zhang[2], Shasha Tu[1], Jingxuan Yang[1], Yu Wang[1], Ruyun Zhou[3], Kui Zhang[2], Jianping Chen[4] and Wenxing Yang[1]

[1] Department of Physiology, West China School of Basic Medical Sciences and Forensic Medicine, Sichuan University, Chengdu, Sichuan, China
[2] Department of Forensic Pathology, West China School of Basic Medical Sciences and Forensic Medicine, Sichuan University, Chengdu, Sichuan, China
[3] Department of Anatomy, Bioimaging and Neuro-cell Science, Jichi Medical University, Tochigi, Japan
[4] Department of Pathogenic Biology, West China School of Basic Medical Sciences and Forensic Medicine, Sichuan University, Chengdu, Sichuan, China
* These authors contributed equally to this work.

Corresponding authors
Jianping Chen,
chenjianping@scu.edu.cn
Wenxing Yang, yangwx@scu.edu.cn

## ABSTRACT

The miniMOS technique has been widely used in the *C. elegans* community to generate single copy insertions. A worm is considered as a potential insertion candidate if it is resistant to G418 antibiotics and does not express a co-injected fluorescence marker. If the expression of the extrachromosomal array is very low, it is possible for a worm to be mistakenly identified as a miniMOS candidate, as this low expression level can still confer resistance to G418 without producing a detectable fluorescence signal from the co-injection marker. This may increase the workload for identifying the insertion locus in the subsequent steps. In the present study, we modified the plasmid platform for miniMOS insertion by incorporating a *myo-2* promoter-driven TagRFP or a ubiquitous H2B::GFP expression cassette into the targeting vector and introducing two loxP sites flanking the selection cassettes. Based on this new miniMOS tool kit, the removable fluorescence reporters can be used to visualize the single copy insertions, greatly reducing insertion locus identification efforts. In our experience, this new platform greatly facilitates the isolation of the miniMOS mutants.

# INTRODUCTION

A transposon is a DNA fragment that can relocate within the genome of a single cell, which is discovered by the Nobel laureate Barbara McClintock (*McClintock, 1950*, *1942*). *Mos1* transposase belongs to the *Tc1/mariner* family, which is relatively conserved among most eukaryotic species and was initially isolated in *Drosophila mauritiana*. It can catalyze the relocation of the transposon (*Hartl, 2001*; *Richardson et al., 2006*). This naturally existing phenomenon, namely transposase-mediated relocation of a transposon, has led to the development of several widely used techniques that help the researchers to modify the
genome of different species in many studies (*Coates et al., 2000*; *Frokjaer-Jensen et al., 2010*, *2008*, *2014*; *Kawakami, 2005*). However, techniques based on *Mos1* transposase could not be applied to *C. elegans* research as *C. elegans* does not naturally harbor *Mos1* elements in their genome until the Jorgensen Lab introduced *Mos1* transposase and elements into *C. elegans* (*Bessereau et al., 2001*). They further applied the *Mos1*-related techniques to generate single copy insertions and deletions, namely mosSCI and mosDEL techniques, respectively (*Frokjaer-Jensen et al., 2010*, *2008*). They also improved the *Mos1* transposon by testing *Mos1* elements with different sizes, and finally identified a minimal *Mos1* element of 550 bp in length with comparable or even better efficiency for *Mos1*-mediated insertion (*Frokjaer-Jensen et al., 2014*). This minimal essential element was named miniMOS thereafter. Nowadays, *Mos1*-related techniques are popular genome editing methods within the *C. elegans* research community.

The pCFJ910 plasmid contains a miniMOS element (*Frokjaer-Jensen et al., 2014*), as shown in Fig 1. We used the plasmid to generate single-copy insertion mutants in our previous projects (*Hao et al., 2018*; *Wu et al., 2019*). The insertion line was initially isolated based on the following criteria: worms were resistant to G418 antibiotics and did not express the co-injection marker. However, we sometimes encountered difficulties in isolating miniMOS lines. It is possible that worms fitting the above criteria may only carry an extrachromosomal array with an extremely low level of plasmids, rather than a single copy insertion. This extrachromosomal array may be sufficient to confer G418 resistance, while its expression of the co-injection marker may be too low to be detected by the microscope. Therefore, we wondered if we can incorporate a visible reporter for miniMOS insertion to increase our chances of isolating mutants with single copy insertion. In this study, we present a modified plasmid tool kit for miniMOS insertion. This kit includes specific fluorescence markers that can be used to visualize single-copy insertions and flanking loxP sites that make the selection cassettes removable. Based on our experience, this new platform greatly facilitates the isolation of strains carrying miniMOS insertions.

## MATERIALS AND METHODS

### Animals

Wildtype strain (N2) of *C. elegans* was obtained from Caenorhabditis Genetics Center (CGC), which was maintained at 20 °C in a 6-cm NGM-agar (nematode growth medium) plate, following previously published protocol (*Brenner, 1974*). Hermaphrodites were used in the study. Except for N2, all other strains in this study were generated by our lab (Table S1). All strains are available upon request.

### Plasmids construction

All newly generated plasmids were constructed by Gibson assembly and sequenced, including pYW347, pYW249, pYW241, pYW242, pYW146, and pYW186. The detailed strategy for molecular cloning and the primers used for PCR are listed in Tables S2 and S3. For constructing pYW241 (Fig. 1), a 2,832 bp fragment upstream of the ATG codon of the *tph-1* gene was used as the *tph-1* promoter. For constructing pYW242 (Fig. 1), *ser-1* promoter and cDNA were amplified from a gift plasmid (*Guo et al., 2018*) from

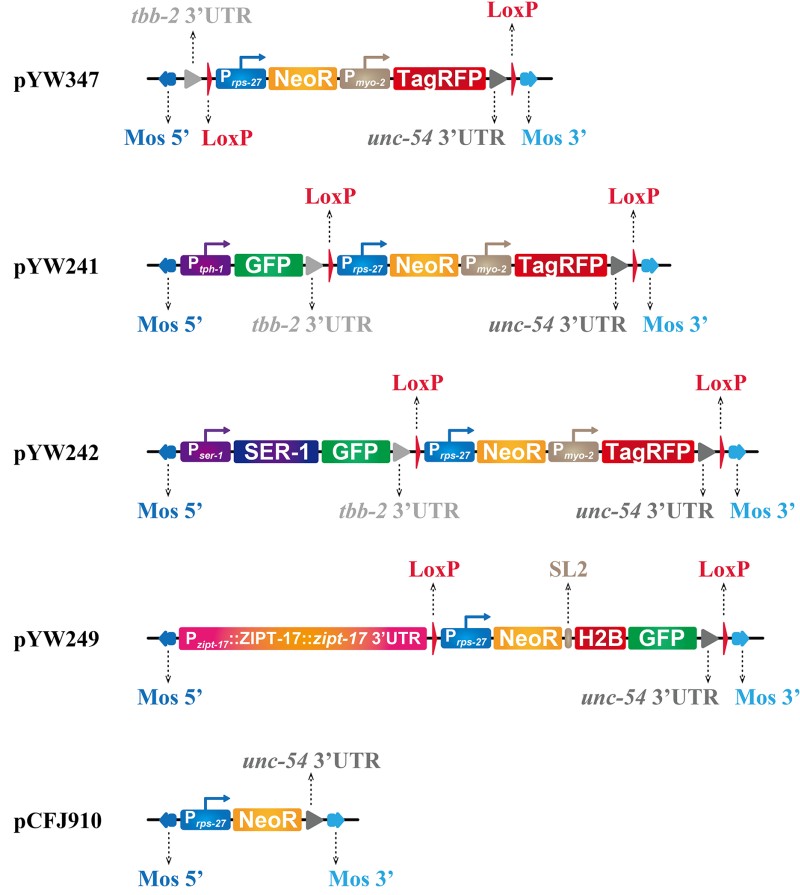

**Figure 1 Plasmids generated to facilitate miniMOS mutant isolation.** All plasmids were constructed using pCFJ910 as an initial template and are ampicillin resistant. The figure shows the relative locations of each element in each plasmid. The length of each fragment in the figure is not proportional to its actual length. The plasmid backbone, containing elements for antibiotic resistance and plasmid replication, is not shown in the figure.

Dr. Taihong Wu (Harvard University, Cambridge, MA, USA). For constructing pYW249 (Figs. 1 and 2), a 5,820 bp fragment covering the whole genomic region of the *zipt-17* gene, starting from −3,914 bp relative to ATG to +503 bp relative to TAA of the *zipt-17* gene, was amplified. For constructing pYW146, a 1,300 bp fragment upstream to the ATG codon of the *lin-44* gene was used as the *lin-44* promoter (*Ge et al., 2020*). The pYW347 and pYW249 plasmids are available in Addgene, with plasmid IDs 196064 and 196063 respectively. All sequences of the above plasmids are listed in Table S2.

All plasmids are available upon request. The authors affirm that all data necessary for confirming the conclusions of the article are present within the article, figures, and tables.

## Microinjection, miniMOS mutant isolation, and insertion locus identification

Microinjection was performed as previously reported (*Rieckher & Tavernarakis, 2017*), using a ZGENEBIO microinjector (ZGene Biotech Inc., Taipei, China), under an inverted microscope. The information on the injection mix is listed in Table S4. All the plasmids

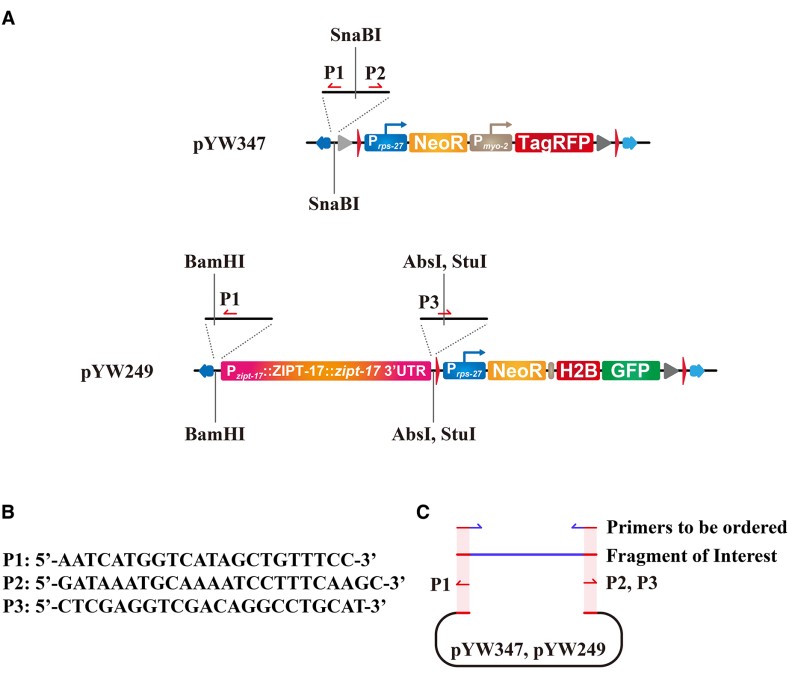

**Figure 2 Molecular cloning strategies for constructing targeting vectors using our plasmid platform.**
(A) Primer locations and directions are indicated by P1, P2, and P3, and restriction enzyme sites for traditional molecular cloning are shown. (B) Sequences of P1, P2, and P3. (C) Gibson assembly strategy for generating targeting vectors. The shaded region in light red shows the homologous region of the primers and the fragments for Gibson assembly. Primers for generating the fragment of interest, indicated as "primers to be ordered", should contain the homologous arm (red) and the sequences for amplification (blue).

were kept at −20 °C before mixing. We used a freshly prepared injection mix for each injection and did not reuse it. All steps from microinjection to mutant isolation can be found in Fig. 3. For identifying the miniMOS locus, inverse PCR was performed according to the protocol reported in the Supplemental Information of the previous publication (*Frokjaer-Jensen et al., 2014*). All PCRs in this study were performed using MonAmp™ 2× Mighty PCR Mix (Monad Biotech Co., Ltd., Shanghai, China) or BeyoFusion™ PCR Master Mix 2× (Beyotime Biotechnology, Jiangsu, China) in an Arhat 96 PCR machine (Monad Biotech Co., Ltd., Shanghai, China). The bands from genotyping PCRs were sequenced to confirm their identities.

## Fluorescence imaging

For Figs. 4A and 4H, the images were taken using an Olympus IX71 fluorescence inverted microscope with a 20× objective. As the original images showed a dim signal, the exposure of these images was adjusted by +4, to facilitate the presentation. For Fig. 4H, five images covering the whole body of the worm were taken separately, and then manually aligned. For taking images in Figs. 4B–4G and 4I–4N, a 6-cm NGM plate was coated with 100 μL of 25 mM potassium azide, then the worm for imaging was picked to the plate and left for free moving. When the worm was immobilized, an image was taken using a MshOt stereo fluorescence microscope MS23, which is composed of an Olympus SZX7 stereo

**1. Preparations before injection**

    (1) plasmid construction by Gibson assembly
    (2) young adult worms for injection
           we usually use day 1 adult worms, carrying about 10 eggs

**2. Injection**

**Day.1.** (1) inject 12 ~ 20 P0 adults
          (2) culture them in NGM plates (injection plate), 3 ~ 5 worms/plate
          (3) kept at 25°C

**3. Start antibiotic screening**

**Day.2.** (1) add G418 antibiotics, 1.25 mg/mL agar
          (2) kept at 25°C, until worms nearly starve

**4. Mutant isolation**

**Day.9.** (1) individually culture potential mutant, 1 worm/plate, **2~4 worms/injection mix**
               **the miniMOS insertion candidate:**
                   • resistant to G418
                   • carries no injection marker
                   • **express dim (but visible) and regularly-distributed $P_{myo-2}$::TagRFT or**
                     **ubiquitous H2B::GFP signals**
          (2) kept at 25°C.

**5. Insertion identification by inverse PCR**

**Day.12 ~ 13.** isolate genomic DNA of the mutants and run inverse PCR
**Day.14 ~ 15.** verify the insertion locus by routine genotyping PCR
**Day.16.** confirm the identity of the PCR bands by sequencing

**6. Isolate the homozygote muntant**

**Day.15.** individually culture the mutant, 1 L4 stage worm/plate, 6 worms per line
               **pick up the worm with regularly-distributed fluorescence signals**
                 (note: homozygote animal may have relatively brighter fluorescence signals)
**Day.19.** verify homozygote
               **the homozygous miniMOS mutant:**
                   • a specific band from genotyping PCR for insertion locus
                   • no specific band from genotyping PCR for wildtype allele
                   • **all progeny carry $P_{myo-2}$::TagRFP or ubiquitous H2B::GFP signals with**
                    **same brightness**

**7. Excision of the indispensible fragments from the genome (optional)**

**Day.21. inject $P_{eft-3}$::Cre plasmid**
               • **culture by 3 worms/injection plate**
               • **kept at 25°C**
**Day.29. isolate animals without $P_{myo-2}$::TagRFP or ubiquitous H2B::GFP signals**
**Day.33. verify the excision by genotyping PCR and sequencing**

**Figure 3 A stepwise protocol of miniMOS animal generation and mutant isolation.** A stepwise protocol shows the details of generating miniMOS nematodes and isolating miniMOS mutants using our new plasmid platform. Literature in red color indicates the specific steps using our plasmid platform.

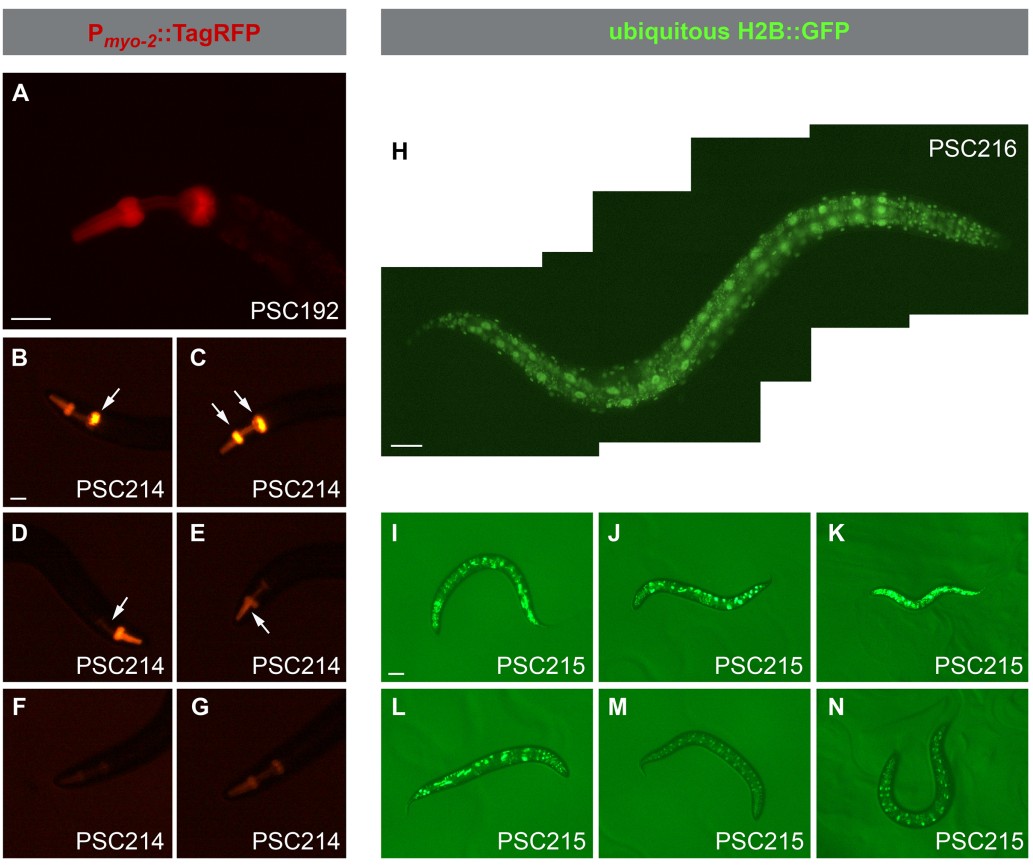

**Figure 4 Expression patterns of different worms with an extrachromosomal array or a single copy insertion.** (A) Fluorescence imaging of a worm with a miniMOS insertion of P$_{myo-2}$::TagRFP. An evenly distributed red signal can be observed restricted to the pharynx. (B–G) Fluorescence imaging of six worms carrying P$_{myo-2}$::TagRFP signals. All worms are progeny of the same mix-injected worms. Among worms labeled (B–E), each may carry an extrachromosomal array, and their P$_{myo-2}$::TagRFP signals are not evenly distributed in their pharynx. The worms labeled (F and G) may have a miniMOS insertion, as their signals are evenly distributed. The fluorescence intensity observed in worms labeled (F and G) is possibly indicative of their genotypes, with (F) possibly being heterozygous and (G) possibly being homozygous. (H) Fluorescence imaging of a worm with a miniMOS insertion of ubiquitous H2B:: GFP. A regularly distributed green signal can be observed all over the body. (I–N) Fluorescence imaging of six worms carrying ubiquitous H2B::GFP signals. All worms are progeny of the same mix-injected worms. Among the worms labeled (I–L), each may carry an extrachromosomal array, and their H2B:: GFP signals are not regularly distributed. The worms labeled (M and N) may have a miniMOS insertion, as their signals are regularly distributed. The fluorescence intensity observed in worms labeled (M and N) is likely indicative of their miniMOS genotypes, with (M) possibly being heterozygous and (N) possibly being homozygous. Strain information is provided in the lower right corner of each figure. Scale bars, 40 μm in (A, B, and H), 80 μm in (I).

fluorescence microscope equipped with an LED fluorescence light source. All images were pseudo-colored according to the fluorescence.

## Quantification of insertion ratio

The quantification of the insertion ratio was performed based on a previously published article with some modifications (*Frokjaer-Jensen et al., 2014*). Briefly, we used two different injection mixes (Table S4). Both mixes contained a miniMos plasmid, a *Mos1* transposase

plasmid, two co-injection markers, and pUC19. One of the mixes contained the negative PEEL-1 selection plasmid. We picked F1 transgenic animals to individual plates and allowed them to starve at 25 °C. These F1 animals were resistant to G418 and carried either $P_{myo-2}$::mCherry or $P_{myo-3}$::mCherry fluorescence, or both. Insertion candidates were initially identified by their G418 resistance and the absence of the injection marker, as previously described (*Frokjaer-Jensen et al., 2014*). Among these candidates, those with regularly expressed fluorescence were the candidates met our criteria. The insertion ratio was calculated by dividing the number of the candidates by the number of F1 animals and multiplying the result by 100%. For peel-1 selection, we heat-shocked the starved plates for 2 h at 34 °C.

## RESULTS

### Modified plasmids for miniMOS insertion

To visualize the insertion, we sought to introduce fluorescent expression cassettes into the pCFJ910 plasmid. The *myo-2* promoter ($P_{myo-2}$) can drive the protein expression exclusively in the pharynx of *C. elegans*, making the *myo-2* promoter-driven fluorescent protein a widely used fluorescent reporter (*Frokjaer-Jensen et al., 2014*; *Semple, Garcia-Verdugo & Lehner, 2010*; *Taylor & Dillin, 2013*; *Toker et al., 2022*). A ubiquitous promoter, such as *rps-27* promoter ($P_{rps-27}$) or *eft-3* promoter ($P_{eft-3}$), driven fluorescent protein fused to histone H2B protein can label the nucleosomes of most cells in *C. elegans* (*Adikes et al., 2020*; *Frokjaer-Jensen et al., 2012*). Both reporters are visible with single copy insertion (*Adikes et al., 2020*; *Frokjaer-Jensen et al., 2012*; *Toker et al., 2022*) and serve as good candidates for our purpose. On the one hand, we used a polyA-trap strategy (*Niwa et al., 1993*) to construct pYW347 plasmid. We introduced $P_{myo-2}$ driven TagRFP, a monomeric red (orange) fluorescent protein (*Shaner et al., 2008*), downstream of the *neoR* gene in pCFJ910 plasmid (pYW347 in Fig. 1). Although the $P_{rps-27}$::neoR cassette lacks an immediate 3′-untranslated region (3′-UTR), it can capture the *unc-54* 3′-UTR downstream of $P_{myo-2}$::TagRFP. To simplify future molecular cloning and avoid repetitive use of the *unc-54* 3′-UTR within one plasmid, we added an extra *tbb-2* terminator upstream of the $P_{rps-27}$ promoter. On the other hand, we fused a green fluorescent protein (GFP) (*Tsien, 1998*) to histone H2B (H2B::GFP) and inserted it downstream of the *neoR* gene in pCFJ910 plasmid, with a spliced leader 2 (SL2) element in between (pYW249 in Fig. 1). Thus, the SL2::H2B::GFP cassette can adopt the $P_{rps-27}$ and *unc-54* 3′-UTR from the plasmid, which drives *neoR* expression in pCFJ910.

Those elements, *e.g.*, TagRFP, NeoR, H2B::GFP, sitting in the targeting vectors, either pCFJ910 or our newly constructed ones, are useful for isolating positive insertion. After that, they become dispensable, or even potentially have a negative influence on the worms. Thus, it is thoughtful if these cassettes can be removed after mutant isolation. Cre-loxP recombination has been widely used for DNA fragment deletion or cassette exchange in different species, including *C. elegans* (*Alemany et al., 2018*; *Hacker et al., 2017*; *Nonet, 2020*; *Siegal & Hartl, 2000*). We inserted two loxP sites, flanking the selection cassette, either $P_{rps-27}$::neoR::$P_{myo-2}$::TagRFP::*unc54* 3′-UTR or $P_{rps-27}$::neoR::SL2::H2B::GFP::*unc54*
3′-UTR, into the plasmids (Fig. 1). Thus, the whole selection cassette is removable by additional injection of *Cre* recombinase expression plasmid.

Based on the newly constructed plasmid platform, we further made three targeting vectors for our ongoing projects: one for showing the *tph-1* gene expression pattern by *tph-1* promoter-driven GFP (pYW241 in Fig. 1), one for labeling *ser-1* gene by GFP (pYW242 in Fig. 1), and one for introducing wildtype *zipt-17* genomic region into the *zipt-17 (ok745)* mutant (pYW249 in Fig. 1). Unfortunately, neither a single copy inserted $P_{tph-1}$::GFP reporter nor $P_{ser-1}$::SER-1::GFP reporter can be seen under the inverted fluorescence microscope.

## Molecular cloning strategy using our new plasmid platform

We recommend using Gibson assembly to do the plasmid construction. Locations of the primers and their sequences were shown in Figs. 2A and 2B. The main backbones of the plasmids, the 7,447 base pair (bp) fragment from pYW347 or the 6,871 bp fragment from pYW249, can be amplified using primer sets P1/P2 or P1/P3, respectively (Fig 2A, Tables S2 and S3). The fragment of interest in your project can be amplified using specific primers carrying 15 to 25 bp homologous arms to the above fragments, as shown in Fig 2C. Alternatively, a traditional molecular cloning strategy based on T4 ligase-mediated DNA ligation can be conducted, using restriction sites SnaBI in pYW347, or BamHI, AbsI, and StuI in pYW249 (Fig. 2A).

## An updated protocol for generating miniMOS line

With our new plasmid platform, we further updated the protocol for isolating miniMOS mutants, based on a previously published protocol (*Frokjaer-Jensen et al., 2014*).

We summarized our protocol in Fig. 3, starting from plasmid construction. Our protocol is composed of seven steps, which usually take about 1 month to generate mutant animals without the indispensable elements for mutant screening, *e.g.*, TagRFP, neoR, H2B::GFP (Fig. 3). Some unique details are reported as follows. We usually use young adult worms carrying about 10 eggs for injection. According to our experience, worms at this age are not as fragile as the younger adult worms and are capable of generating more progeny than the older ones. With the help of our fluorescence expression cassettes, $P_{myo-2}$::TagRFP or ubiquitous H2B::GFP, the insertion candidates can be primarily identified by the expression of dim and regularly-distributed fluorescence in the pharynx or all over the body of the worms (Fig. 3 step 4, and Fig. 4). Because of the existence of a single copy insertion of fluorescence reporter, the homozygote candidate worm may be isolated by picking up worms with a relatively brighter fluorescence signal (Fig. 3, step 6). These visible reporters are also useful for further verifying the homozygous animals, whose progeny should all carry fluorescent reporters (Fig. 3, step 6). An optional step for excising these indispensable elements for mutant screening can be conducted by injection of $P_{eft-3}$::Cre plasmid (Fig. 3, step 7).

## Facilitated isolation of miniMOS candidate by the fluorescence markers

Here we present our results and criteria for the isolation of insertion candidates based on fluorescence reporters, *i.e.*, $P_{myo-2}$::TagRFP or ubiquitous H2B::GFP. The signals from the $P_{myo-2}$::TagRFP cassette were restricted in the pharynx of the *C. elegans* (Figs. 4A–4G), which is consistent with previous publications (*Frokjaer-Jensen et al., 2014*; *Semple, Garcia-Verdugo & Lehner, 2010*; *Taylor & Dillin, 2013*; *Toker et al., 2022*). The signal from a single copy inserted $P_{myo-2}$::TagRFP cassette was smoothly distributed in the pharynx, they were not so bright, and nearly symmetrical along the anterior-posterior axis of the pharynx, especially in the pharyngeal bulbs of the worm (Fig. 4A). After the injection, the non-transgenic animals, which carried no extrachromosomal array and single copy insertion, were killed by G418 antibiotics. More than 95% of the surviving animals carried an extrachromosomal array. The expression pattern of extrachromosomal $P_{myo-2}$::TagRFP cassette varied among worms (Figs. 4B–4E), which we refer to as irregular patterns hereafter in this article. The irregular patterns of the $P_{myo-2}$::TagRFP cassette expression are diverse, they can be saturated fluorescent signals in one or two pharyngeal bulbs (Figs. 4B and 4C), or hardly visible in the isthmus and terminal bulb of the pharynx (Fig. 4D), or absent from half of the procorpus of the pharynx (Fig. 4E). The signals of the $P_{myo-2}$::TagRFP cassette in these worms were not evenly distributed in the pharynx. In addition to these irregular patterns, we also observed two regular patterns, with the red fluorescence signals distributed smoothly in the pharynx at different strengths (Figs. 4F–4G).

We speculated that the intensities of the signals in these regular patterns possibly indicate the genotype of the insertion locus, or may be caused by the position effect variegation as a result of miniMOS insertions at different loci (*Frokjaer-Jensen et al., 2016*). In our plasmid platform, we included two loxP sites flanking the fluorescence reporter (Fig. 1). To test if the fluorescence reporter is removable, we injected the worms with a *Cre* recombinase-expressing plasmid and picked their progeny without $P_{myo-2}$::TagRFP fluorescence as candidates for genotype analysis. A specific PCR band confirmed the genotypes of the loxP-fragment-deleted locus (Fig. S1).

In our plasmid platform, the H2B::GFP fusion gene was driven by the $P_{rps-27}$ promoter, which led to the ubiquitous labeling of nuclei of the worms (Figs. 4H–4N). The signals from a single copy-inserted $P_{rps-27}$::H2B::GFP cassette gave the worms with a distinct regular pattern (Fig. 4H), consistent with previous publications (*Adikes et al., 2020*; *Frokjaer-Jensen et al., 2012*). Similar to our observation in single copy-inserted $P_{myo-2}$::TagRFP signals, these signals were also not so bright (Figs. 4H, 4M and 4N).

The extrachromosomal $P_{rps-27}$::H2B::GFP showed saturated signals, its expression pattern was quite different from that of the single copy-inserted one and seemed to follow no regularity (Figs. 4I–4L). We also observed two regular patterns carrying green fluorescence signals with different strengths (Figs. 4M and 4N), which may result from different genotypes or position effect variegation. The loxP-flanked H2B::GFP fragment can be deleted by injecting the worms with a *Cre* recombinase-expressing plasmid, and its genotype was confirmed by a specific PCR band (Fig. S1).

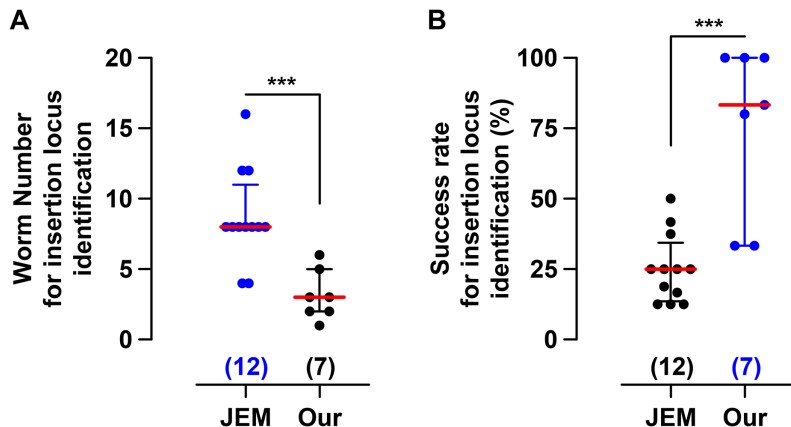

**Figure 5 Our protocol facilitates the isolation of the miniMOS insertion locus.** (A) The number of worms used for inverse PCR. (B) Success rate of insertion locus isolation. Each dot represents an individual injection for generating miniMOS insertion. The blue color indicates the data are not normally distributed. The numbers above the x-axis denote the sample scale of the corresponding group. The bars show the median with the interquartile range. ***$p < 0.001$, by Mann-Whitney test to compare ranks.

## Facilitated characterization of insertion locus

To test whether our method is more efficient, we compared our method with a previously published one by the Jorgensen EM group, which is referred to as the JEM method or protocol hereinafter. From December 2020 to November 2022, we injected a total of 19 plasmid mixtures for different ongoing projects in our lab, 12 using the JEM method and seven using our protocol. Firstly, we compared the number of worms used for insertion locus identification by inverse PCR. On average, the JEM method used about seven worms per injection mix, and our method used about three worms per injection mix (Fig. 5A). Considering each worm would be used for an independent inverse PCR experiment to identify the insertion site, this result indicates that our method is less labor-intensive. Secondly, we compared the success rate in insertion locus identification using these two methods, which was 23.8% for the JEM method and 75.7% for our method on average (Fig. 5B). This result suggests that our success rate in identifying the insertion site is almost three times as efficient as the JEM method. Furthermore, we also compared the insertion efficiency using injection mixtures with or without PEEL-1 selection (Table S5).

We observed that the PEEL-1 selection by heat shock killed the transgenic worms in most cases, but we didn't find a significant difference in insertion ratios between the injections with and without PEEL-1 selection (Table S5, 10.1% *vs* 9.7% by JEM method, or 6.8% *vs* 6.2% by our method). We also noticed a tendency of our protocol to yield a comparatively lower insertion rate, compared to the JEM method (Table S5, 10.1% *vs* 6.8% for injections without peel-1 selection, or 9.7% *vs* 6.2% for injections with PEEL-1 selection). However, even with this potentially lower insertion ratio, our method still had a very high efficiency, about 75.7%, in insertion locus identification. Therefore, the above data collectively suggest that our method for generating miniMOS insertion lines is more effective for insertion identification.

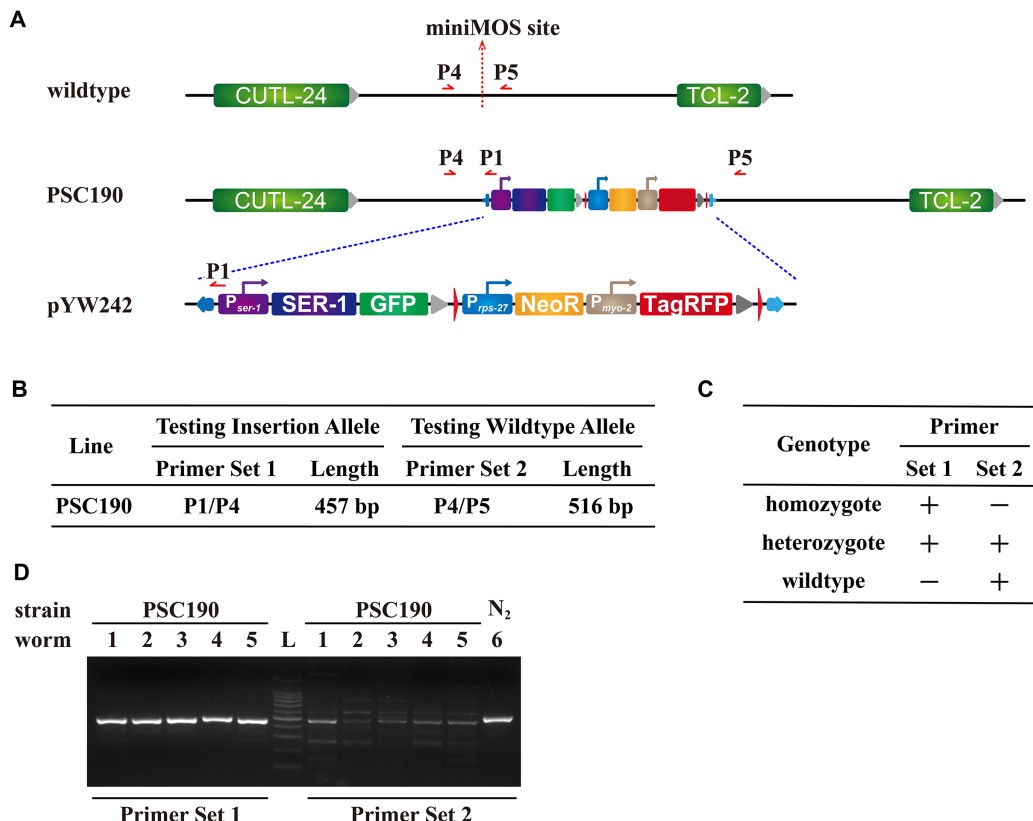

**Figure 6 Genotyping strategies and results for miniMOS lines.** (A) Graphic illustration of the insertion locus in the PSC190 miniMOS line. The red arrows indicate the locations and directions of the genotyping primers P1, P4, and P5. (B) Strategies of genotyping PCR. (C) Result interpretation strategy based on genotyping PCR. Note that the PCR extension time is set to be just enough for generating bands shorter than 1 kb, so PCR using P4 and P5 should not be able to generate specific bands covering the whole insertion locus in PSC190. (D) PCR results for PSC190. L represents DNA Ladder, indicating 100, 200, 300, 400, 500, 600, 700, 800, 900, 1,000, and 1,500 bp, respectively. The identity of the PCR band was confirmed by sequencing. According to sequencing results, the bands generated by primer set 2 using PSC190 genomic DNA as a PCR template are all nonspecific.

## The fluorescence reporter facilitates the isolation of homozygote animals

If all the progeny of the candidate worm are fluorescent and have similar expression patterns as the candidate worm, they are likely homozygotes. Our newly generated strain PSC190 met this criterion. We performed a series of genotyping PCRs to further verify their genotypes. A correct PCR band from primer set 1 suggests the presence of the insertion, and that from primer set 2 suggests the presence of the wildtype allele (Figs. 6A–6C). Genotyping PCR of five PSC190 worms with primer set 1 all generated a clear band, indicating insertion (Fig. 6D). In contrast, genotyping PCR of these worms with primer set 2 generated a series of bands. The band with a similar length to the positive control was confirmed to be nonspecific by sequencing, indicating the absence of wildtype locus in PSC190 (Fig. 6D). Since the progeny of PSC190 are 100% fluorescent, the above results collectively suggest that these randomly picked worms were all homozygotes.

## DISCUSSION

### Single copy inserted fluorescence reporters facilitate the identification of miniMOS mutants

In the present study, we established a new platform for generating miniMOS insertions, by incorporating fluorescent expression cassettes into the targeting vector pCFJ910. This new platform is composed of two plasmids, pYW347 and pYW249, using $P_{myo-2}$::TagRFP or ubiquitous H2B::GFP as a visible reporter, respectively.

Our new miniMOS platform has several advantages. Firstly, these fluorescence reporters can greatly facilitate miniMOS mutant isolation. As we presented in Fig. 4, the miniMOS candidate lines generated by our plasmid platform have regular expression patterns, compared to the extrachromosomal array-containing lines. These expression patterns were easily recognizable. This visible reporter facilitated the isolation of the candidate worms. Secondly, the insertion locus identification using our protocol is less labor-consuming. The number of worms used for the inverse PCR step is reduced by more than 50%, and the success rate for identifying the insertion locus is about 75%, which is nearly tripled. These findings suggest that our method greatly reduced the labor for identifying the insertion site. Thirdly, with the aid of the fluorescent reporter, homozygote isolation can be easily confirmed by whether their progeny are 100% fluorescent, which is more straightforward and saves some efforts from genotyping PCR, thus facilitating homozygote isolation.

We also acknowledge some limitations of our platform. Firstly, the fluorescence cassettes are 1.4 to 1.7 kb in length, which increase the length of the final plasmids. This may potentially increase the difficulty of molecular cloning. Secondly, even if the selection cassettes are removed, one loxP site would permanently remain in the genome. Although the impact of one loxP site on animal physiology might be tiny, it can be troublesome when you cross this worm into another genomic background carrying both loxP (or its mutant) sites and *Cre* recombinase. Thirdly, $P_{myo-2}$ promoter-driven fluorescence marker can be lethal when a higher concentration of this plasmid exists in the injection mixture (*Rieckher & Tavernarakis, 2017*). When you use pYW347-based plasmid as a template for your miniMOS work, we recommend a working concentration lower than 17 ng/μL. Finally, according to our experience, the ubiquitous H2B::GFP signals are more sensitive to our eyes than $P_{myo-2}$::TagRFP signals under the stereo fluorescence microscope, when searching for the candidate worms from a mixed population containing miniMOS worms and transgenic worms; while the loss of the $P_{myo-2}$::TagRFP signals is more sensitive to our eyes when searching for worms without fluorescence. Thus, which is better depends on whether you plan to remove the loxP-flanked fluorescence cassettes.

## ACKNOWLEDGEMENTS

We thank Prof. Di Chen (Nanjing University) for sharing the pCFJ601 plasmid, and Dr. Taihong Wu (Harvard University) for sharing the plasmid containing the *ser-1* promoter and *ser-1* cDNA sequence.

### Funding

Wenxing Yang received funding from the Fundamental Research Funds for the Central Universities (China) and the National Natural Science Foundation of China (No. 32271178). Kui Zhang received funding from the Sichuan Provincial Department of Science and Technology (China). The funders had no role in study design, data collection and analysis, decision to publish, or preparation of the manuscript.

### Grant Disclosures

The following grant information was disclosed by the authors:
Fundamental Research Funds for the Central Universities (China).
National Natural Science Foundation of China: 32271178.
Sichuan Provincial Department of Science and Technology (China).

### Competing Interests

The authors declare that they have no competing interests.

### Author Contributions

- Jiangyun Li performed the experiments, analyzed the data, prepared figures and/or tables, authored or reviewed drafts of the article, and approved the final draft.
- Yuang Qin performed the experiments, analyzed the data, prepared figures and/or tables, authored or reviewed drafts of the article, and approved the final draft.
- Chengchen Shen performed the experiments, analyzed the data, prepared figures and/or tables, authored or reviewed drafts of the article, and approved the final draft.
- Jun Zhang performed the experiments, authored or reviewed drafts of the article, and approved the final draft.
- Shasha Tu performed the experiments, authored or reviewed drafts of the article, and approved the final draft.
- Jingxuan Yang performed the experiments, authored or reviewed drafts of the article, and approved the final draft.
- Yu Wang performed the experiments, authored or reviewed drafts of the article, and approved the final draft.
- Ruyun Zhou conceived and designed the experiments, authored or reviewed drafts of the article, and approved the final draft.
- Kui Zhang conceived and designed the experiments, authored or reviewed drafts of the article, and approved the final draft.
- Jianping Chen conceived and designed the experiments, authored or reviewed drafts of the article, and approved the final draft.
- Wenxing Yang conceived and designed the experiments, prepared figures and/or tables, authored or reviewed drafts of the article, and approved the final draft.

## Data Availability

All the data is available in the figures and the Supplemental Files.

## Supplemental Information

Supplemental information for this article can be found online at http://dx.doi.org/10.7717/peerj.15433#supplemental-information.

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
