# Peer review of "A new miniMOS tool kit capable of visualizing single copy insertion in C. elegans"

_PeerJ, doi:10.7717/peerj.15433_

## Round 0.1 · original submission · Major Revisions

While there appears to be split decisions among reviewers, two of the three reviewers ask for better quantification of the data and both suggest including the sequence of the vector. Critically, reviewer 1 pointed out the claims that the vectors are freely available at AddGene were not substantiated at the time of review. This reviewer also provides critical correction to terminology and important technical controls that much be answered to substantiate this as an advancement in protocol.

Reviewer 1 ·

Basic reporting

The manuscript by Li et al describe an expanded toolkit for miniMos transgenesis in C. elegans. The main improvements are: (1) the inclusion of fluorescent markers inside the miniMos transposon, which allows easier identification of single-copy insertions, (2) the inclusion of LoxP sites flanking the selection markers (the fluorophore and the antibiotic selection marker, NeoR), and (3) inclusion of a SnaBI cloning site to insert the transgene of interest. The authors suggest that these adaptions, together with a detailed protocol, have facilitated transgenesis in the author's laboratory, in particular the identification of homozygous and heterozygous insertion animals.

Although these improvements may be useful, I have some concerns and suggestions for improvements to the manuscript that I will detail in the following sections.

The manuscript has grammatical issues that can be fixed in the copy-editing process. This would likely require a fluent English speaker to read through and comment on errors but that is beyond the scope of a reviewer.

The manuscript covers sufficient details of the literature background but appears to mix up some of the terminology commonly used in the field. Generally, miniMos insertions are not referred to as "mosSCI" insertions because the insertion site is random. I would suggest changing the nomenclature throughout.

The manuscript is structured in a reasonable and professional way, consistent with professional standards. However, the raw data makes it difficult to fully understand the modified vector design. The authors indicate that two of the vectors are available for distribution at Addgene, but they have not been made public, and the sequences are not included in the paper. Thus, it is not possible to understand some of the changes made. This is a particular problem because the description of one of the vectors suggests an unusual change (see below). I would encourage the authors to include a vector sequence in the supplementary materials.

The overall hypothesis and the flow of the paper is confusing and could be substantially improved. For example, in the abstract, the authors emphasize how the improved vectors can minimize the effect of false positive insertions due to low expression from co-injected fluorescent plasmids. But in the results, the authors use these fluorescent markers to differentiate between heterozygous and homozygous insertions.

Experimental design

The authors could substantially improve their experiment design to substantiate the utility of the vector kit. The rationale behind the vectors is clear but the authors do not test many of the suggested improvements. For example, they do not test if their fluorescent markers minimize the number of false positive insertions. Nor do they compare their efficiency of insertion with standard protocols in the field, where false positives are generally not a problem if three fluorescent co-injection markers are included. Thus, the reader is left with an impression that the authors solve a problem that is not clearly defined. As a second example, the authors include LoxP sites to enable excision of the fluorescent and antibiotic selection marker. The rationale for this is clear and should be useful. However, the authors do not test if excision after injection of Cre recombinase actually works. As a third example of slightly confusing experimental design, the authors show several plasmids in Figure 1 that do not appear to be used in the manuscript. There are no images of Pser-1::ser-1::gfp or Pzipt-17::zipt-17 and their inclusion in the figure is unclear/confusing.

Furthermore, the design of pYW241 is counterintuitive. The authors insert the Pmyo-2::tagRFP cassette into the NeoR cassette without including a second 3' UTR. In this case, the neoR cassette does not have a 3' UTR which would be expected to make expression of NeoR unstable. There may be a cryptic terminator and polyA sequence in Pmyo-2 but the authors should justify this transgene design.

Validity of the findings

Unfortunately, the results shown by the authors are largely anecdotal and qualitative. For transgenic methodologies, I would expect the authors to test and quantify any improvements to the vector kit they developed. As mentioned in the prior section, most of the suggested improvements are not tested:
1. Do the vectors minimize the number of false positives relative to the standard protocol?
2. What is the insertion frequency? Based on the protocol and the number of injected animals placed on every plate, the insertion frequency appears to be relatively low but is not quantified anywhere.
3. Can the selection marker be excised by Cre recombinase injection?
4. How robust is the selection for heterozygotes versus homozygous inserts? The authors demonstrate just two insertions, but this constitutes a very small sample size. Also, the use of PCR to determine homozygous versus heterozygous insertions is a strange choice since the fluorescent segregation pattern would show this simply by scoring the animals. The use of the absence of a band in a PCR reaction is a tenuous assay that I would recommend against (Figure 5E) as spurious PCR bands will be highly sensitive to contamination. The gel images demonstrate this with several dim bands at the expected size.

Additional comments

Line 106-107. The plasmids are not currently available.

Line 151-152. Can the authors please explain what they mean by "self-excision"? There are, to my knowledge, no reports of excision of transgenes that have two repeats. And based on the number of stable repeats in the genome, I would not expect a repeat to be a problem.
There may be concerns about an HR repair process but since the mos1 transposon moves by "cut and paste" this should also not be a problem.

Line 220-222. Strength of signal. The signal strength could indicate homozygotes versus heterozygotes, as suggested, but could just as well indicate position effect variegation (PEV). It is well known that the insertion site can influence the level of expression (see, for example, Froekjaer-Jensen et al., Cell, 2016). Based on N = 2, it is difficult to generalize their observations.

I believe the improved reagents might well be useful for other laboratories, but the data shown in the manuscript do not rise to the level of conclusively demonstrating this.

Reviewer 2 ·

Basic reporting

I thank the authors for a detailed explanation of their new toolkit. This paper from Li et al. in Yang lab is showing a clever improvement on mosSCI technique to generate single copy transgene insertions in C. elegans.
I liked the level of information provided in the introduction. I do notice a few points missing though. I would like to see authors touch on CRISPR and explain how mosSCI can be advantageous in the light of recent CRISPR developments. Another point that is missing is that the authors do not talk about using peel-1 as a negative selection against formation of extrachromosomal arrays as described in Frokjaer-jensen et al. Nat Methods 2012. It would be a great addition to the introduction and discussion. A comparison of peel-1 selection to their method would be an addition to the manuscript that would benefit the worm community.
I found the language to be clear and engaging, I thank the authors for it. I have a few minor points that I think need improvements in the statements for more clarity. I outline them below.
Specific points:
Title: I think "the" should be removed.
Line 35: ..reporters can be used to visualize..
Line 54: I don’t think University of Utah is necessary.
Line 111: If you are planning on submitting strains to CGC, mentioning it as well will be great.
Line 149: 3’-untranslated region
Line 187: needs rephrasing
Line 190: Using "young" adults instead of "day1" adults might be easier to explain.
Line 215: Instead of using the term “overwhelming majority”, numbers will be great.
Line 261: 5 worms out of how many? Percentages will be helpful.
Line 297: You start talking about third and fourth advantages of this technique but earlier you mentioned your reasons are two-fold. This part would need rephrasing.
Line 310: It’ll be great to mention these concentrations in the methods as well.

Figures are clear and relevant. One minor correction I’d like to see is the addition of scale bars to worm images in Figure 4.

I thank the authors for submitting raw data. One additional piece of information that would be useful is the full sequence of the plasmids they generated.

Experimental design

I thank the authors for a detailed explanation of their new toolkit. The experiments and the figures clearly show that the technique is working well to select correct integration events in C. elegans using mosSCI. Adding how well this method works compared to peel-1 selection would convince me even more. This would answer the question of how well this toolkit improves mosSCI selections.
In order to have a rigorous assessment of the technique developed, I would encourage the authors to add numbers to their data. I would like to see n numbers and percentages on the images they show and also for the genotypings they do. For eg. 5/how many worms showed homozygotes versus heterozygotes in their hands.
With respect to the methods, I would like details on concentrations of the plasmids used when making injection mixes to be included in the microinjection method details. How the injection mix is stored and for how long is another detail I would appreciate.

Validity of the findings

I like that the conclusions do not include overstatements and clearly outline the caveats. Though there are no biological questions addressed, I find the method as an improvement to mosSCI that would benefit the C. elegans community. The four figures of the manuscript show in detail how this method works, and leave me entirely convinced that this method provides another convenient means to screen through insertions in worms.
The manuscript would benefit from tightening up the results section with more quantitative numbers and comparisons to established mosSCI selections by showing how many less worms would have to be screened to catch an integration event with their method versus others.

In summary, I find the technique to be elegant, but the biological insights currently achieved with this strategy to be limited. Thus, it’d be a nice methods paper I’d be willing to accept.

Additional comments

No additional comments. Thanks again for this new finding.

·

Basic reporting

The manuscript is written very clearly by the authors. There are just minor grammatical issues that should be corrected to improve the readability of the manuscript (for example sentences starting on lines 44, 61 and 63). I would suggest the authors review the manuscript for these grammatical errors,

In terms of content, the authors provide sufficient background and context for the improvements they designed on the mosSCI protocol and the figures are all well made.

Experimental design

I have no comments on the experimental design. The authors clearly lay out the potential pitfall of standard mosSCI protocol not identifying true insertions.

Validity of the findings

The authors' suggested enhancement of the MosSCI protocol is well laid out, and the data provided supports the utility of additional fluorescent reporters.

---

## Round 0.2 · accepted · Accept

Thank you for addressing the concerns of the reviewers. They are pleased with the resubmission and I am delighted to accept the manuscript for publication.

Reviewer 1 ·

Basic reporting

The structure of the manuscript is greatly improved

Experimental design

All major experimental concerns have been addressed.

Validity of the findings

The updated manuscript now includes the necessary information to assess the findings, which are validated.

Additional comments

I would like to commend the authors for the thorough revision and improvements to the manuscript.

·

Basic reporting

The authors have fixed the grammar and language throughout the article in this revision and greatly improved the readability of the article.

Experimental design

no comment

Validity of the findings

no comment

Additional comments

The article seems ready for acceptance.